# The Effect of Quinolones on Common Duckweed *Lemna minor* L., a Hydrophyte Bioindicator of Environmental Pollution

**DOI:** 10.3390/ijerph20065089

**Published:** 2023-03-14

**Authors:** Łukasz Sikorski, Agnieszka Bęś, Kazimierz Warmiński

**Affiliations:** Department of Chemistry, Faculty of Agriculture and Forestry, University of Warmia and Mazury in Olsztyn, Pl. Łódzki 4, 10-727 Olsztyn, Poland

**Keywords:** antibiotics, quinolones, aquatic plant, biosorption, phytotoxicity, photosynthesis

## Abstract

Plant growth and the development of morphological traits in plants are inhibited under exposure to pharmaceuticals that are present in soil and water. The present study revealed that moxifloxacin (MOXI), nalidixic acid (NAL), levofloxacin (LVF) and pefloxacin (PEF) at concentrations of >0.29, >0.48, >0.62 and >1.45 mg × L^−1^, respectively, inhibited the growth (Ir) of duckweed plants and decreased their yield (Iy). In the current study, none of the tested quinolones (QNs) at any of the examined concentrations were lethal for common duckweed plants. However, at the highest concentration (12.8 mg × L^−1^), LVF increased Ir and Iy values by 82% on average and increased the values of NAL, PEF and MOXI by 62% on average. All tested QNs led to the loss of assimilation pigments. In consequence, all QNs, except for LVF, induced changes in chlorophyll fluorescence (Fv/Fm), without any effect on phaeophytinization quotient (PQ) values. The uptake of NAL, MOXI, LVF by *Lemna minor* during the 7-day chronic toxicity test was directly proportional to drug concentrations in the growth medium. Nalidixic acid was absorbed in the largest quantities, whereas in the group of fluoroquinolones (FQNs), MOXI, LVF and PEF were less effectively absorbed by common duckweed. This study demonstrated that biosorption by *L. minor* occurs regardless of the plants’ condition. These findings indicate that *L. minor* can be used as an effective biological method to remove QNs from wastewater and water and that biosorption should be a mandatory process in conventional water and wastewater treatment.

## 1. Introduction

Aquatic and terrestrial environments are contaminated with numerous xenobiotics that are widely used in households, agriculture and industry and are evacuated with wastewater to the environment [1]. Pharmaceuticals, including antibiotics, are an important category of xenobiotics. In the modern world, antibiotics are used extensively in human and veterinary medicine, agriculture, livestock farming, fish farming and aquaculture [2]. According to the World Health Organization (WHO), the animal sector is responsible for approximately 80% of total consumption of medically important antibiotics [3]. Each year, more than 100,000 tons of antibiotics are consumed worldwide, which illustrates the scale of the problem [4]. Antibiotics that are applied directly to human and animal tissues are not fully metabolized, they are partially excreted with urine and feces and are transported to terrestrial and aquatic environments with wastewater, manure and slurry. Antibiotics present in the environment are highly toxic to humans, animals and plants [5,6]. Municipal, hospital and industrial wastewater is the key source of pharmaceuticals that reach the environment through surface runoffs, percolation to groundwater and fertilization [7,8,9,10]. Numerous studies have shown that more than 16 popular classes of antibiotics, from β-lactams to macrolides, are ubiquitous in wastewater treatment plants (WWTPs) and the aquatic environment [11,12,13,14,15,16,17,18,19,20]. Antibiotics evacuated to water bodies with treated wastewater pose a significant threat to human health with the arrival of bacteria resistant to antibacterial drugs and by changing the metabolism of these organisms as well as the environment. Therefore, effective methods of removing these pollutants from wastewater need to be developed [21]. Untreated effluents from industrial plants contaminate drinking water, which is highly toxic to living organisms, causing, among other resistance mutations, the emergence of plasmid-mediated resistance, even when water contains traces amounts of antimicrobials in the range of ng × L^−1^ [22,23]. Among the antimicrobial substances, there is a group of synthetic chemotherapeutic drugs—quinolones (QNs).

QNs are used in the treatment of infectious diseases in human medicine [24,25]. Nalidixic acid was the first synthetic QN antibiotic designed to treat urinary tract infections caused by Gram-negative bacteria. However, due to growing levels of bacterial resistance to QNs and environmental contamination with these drugs, NAL has been gradually replaced with new, more effective antimicrobials [26,27]. Fluoroquinolones (FQNs) such as PEF, LVF and MOXI were introduced to human and veterinary medicine as well as to livestock production [28,29,30,31,32]. As a result, FQNs have become ubiquitous in surface water, leading to the contamination of the euphotic zone, which plays a key role in ecosystem stability [33]. FQNs are particularly high in environments that are strongly contaminated with wastewater from pharmaceutical plants. In industrial canals in central-southern India, MOXI concentration reached an alarming high of 694.1 ng × L^−1^ [34]. Wastewater treatment plants remove organic matter as well as mineral and organic compounds, but they are not highly effective in eliminating pharmaceuticals from wastewater [35]. Only substances that are specified in wastewater discharge guidelines are removed in WWTPs, and recycled wastewater contains pharmaceutical compounds [35]. Due to their complex chemical structure, pharmaceuticals are highly resistant to degradation, and they can persist in the environment for long periods. For this reason, alternative solutions involving biological methods are being sought after to neutralize and minimize the harmful effects of pharmaceuticals in aquatic ecosystems and the plants present there, which are: habitat, a place of organisms, source of oxygen, and primary producers [36]. One of the methods of removing pharmaceuticals from water is biosorption, a physicochemical process during which contaminating substances are removed from a solution by biological materials. Biosorption can be either an active or a passive process that binds pollutants in the cell structure. Plants absorb pharmaceutical compounds, including antibiotics, from water and soil and distribute them throughout plant tissues by passive transport with the transpiration stream [37]. In general, biosorption represents all interactions between an adsorbate and a biological matrix. This process can occur in dead cell and tissue fragments, but it can also be induced by living organisms, where adsorption takes place by surface complexation. During surface complexation, the adsorbate is deposited on cell walls and external cell layers. This process is the first and reversible step of adsorption that proceeds much more rapidly than the general and more complex bioaccumulation mechanism [38].

The following quinolone antibiotics were analyzed in this study:

First generation QNs—nalidixic acid (NAL)—naphthyridine derivative;

Second generation QNs—pefloxacin (PEF)—fluoroquinolone (FQN);

Third generation QNs—levofloxacin (LVF)—FQN;

Fourth generation QNs—moxifloxacin (MOXI)—FQN.

The aim of this study was to determine the effects of exposure to four antibiotics: NAL, PEF, LVF and MOXI, popular drugs in common duckweed (*L. minor*), which is an important link in the food chain in freshwater environments (ISO, 2005; OECD, 2006). The current assessment of toxicity is based mainly on the analysis of common duckweed growth parameters: growth rate, yield and weight. However, they do not include research on the assessment of plant productivity at the biochemical level, after exposure to toxic substances, as well as their specific response mechanisms [39], such as those included in this study. In our study, the phytotoxicity of the examined antimicrobials was determined not only by evaluating the morphological but also the physiological parameters of common duckweed with the aim of establishing common duckweed’s potential as a tool for removing these pharmaceuticals from wastewater and water.

## 2. Materials and Methods

### 2.1. Plant Biosorbent

Common duckweed (*Lemna minor* L.) plants used in the study were obtained from the collection of the Department of Chemistry of the University of Warmia and Mazury in Olsztyn, Poland.

### 2.2. Chemical Adsorbates

Antibiotics:

Nalidixic acid, PEF, LVF and MOXI belonging respectively to the 1st, 2nd, 3rd, 4th generation of QNs were purchased from Sigma Aldrich, Saint Louis, Missouri, USA (Table 1).

### 2.3. Lemna Test

Common duckweed was grown in 50 mL of OECD medium for testing chemicals [40] in a plant growth chamber (ALL–Round–Al 185–4, Gent, Belgium) illuminated with fluorescent light (140 μmol photon m^−2^ × s^−1^ PAR) in a 16 h light/8 h dark cycle (mean maximum temperature of 20 °C during daytime and 16 °C during night time) for 7 days. The biotest was validated in accordance with OECD guidelines: in the temperature 24 ± 2 °C and pH at 6.5 ± 0.2, the doubling time of frond number in the control was less than 2.5 days (60 h), corresponding to approximately a seven-fold increase in seven days and an average specific growth rate of 0.275 d^−1^ [40]. All solutions were prepared using deionized water (Adrona Crystal 5 Basic water purification system, Riga, Latvia) with analytically pure NAL, PEF, LVF and MOXI. The responses of common duckweed to all concentrations of the tested solutions (0; 0.4; 0.8; 1.3; 3.2; 6.4; 12.8 mg × L^−1^) were determined based on the percent inhibition of growth rate (Ir), percent reduction in yield (Iy), percent reduction in dry weight (I_DW_), Chl *a* and Chl *b*, TCC, PQ of tissues, Fv/Fm, and the fresh (FM) and dry matter content (DM) of 100 duckweed plants. Frond area was measured using the Lucia 5.0 program (Laboratory Imaging, s.r.o., Prague, Czech Republic). The following formulas were applied to calculate the values of Ir and Iy based on the number of duckweed fronds and frond area according to OECD guidelines [40], and the values of I_DW_:μ_i–j_ = [ln(N_j_) − ln(N_i_)]/t(1)
Ir = (μ_c_ − μ_T_)/μ_c_ × 100 (2)
Iy = (b_c_ − b_T_)/b_c_ × 100 (3)
where:μ_i–j_—average specific growth rate in time i to j;N_i_—measurement variable in the test or control vessel at time i;N_j_—measurement variable in the test or control vessel at time j;t—time from i to j;μ_c_—mean value of µ in the control group;μ_T_—mean value of µ in the treatment group;b_c_—final number of duckweed fronds and frond area minus the initial number of duckweed fronds and frond area in the control group;b_T_—final number of duckweed fronds and frond area minus the initial number of duckweed fronds and frond area in the treatment group.
%I_DW_ = (DW_c_ − DW_t_)/DW_c_ × 100 (4)
where:%I_DW—_percent reduction in dry weight;DW_c_—dry weight of duckweed fronds in the control group (mg);DW_t_—dry weight of duckweed fronds in the treatment group (mg).

### 2.4. Chlorophyll, Total Carotenoid Content and Phaeophytinization Quotient

The contents of Chl *a*, Chl *b* and TCC were determined in leaf extracts prepared with 96% (*v*/*v*) aqueous ethanol and centrifuged at 12,000× *g* for 15 min. The supernatant was separated; the pigments were quantified spectrophotometrically (Hitachi U-1800 spectrophotometer, Tokyo, Japan) according to [41], and the PQ was calculated according to [42]. The following formulas were used to calculate the concentrations of Chl *a* and Chl *b*, TCC and PQ:Chl *a* = C_Chl_ *_a_* × V_e_ × x/m (5)
Chl *b* = C_Chl_ *_b_* × V_e_ × x/m (6)
C_Chl *a*_ = 13.95 × A_665_ − 6.88 × A_649_(7)
C_Chl_ *_b_* = 24.96 × A_649_ − 7.32 × A_665_
(8)
TCC = [1000 × A_470_ − 2.05 × Chl *a* − 114.8 × Chl *b*]/245 (9)
PQ = A_435_/A_415_
(10)
where:

Chl *a*, Chl *b*—chlorophyll content in plant material (µg × mg^−1^);

V_e_—volume of ethanol extract (mL);

x—dilution coefficient;

m—sample weight (mg);

C_Chl_ *_a_*—concentration of Chl *a* in the extract (µg × mL^−1^);

C_Chl_ *_b_*—concentration of Chl *b* in the extract (µg × mL^−1^);

A_i_—solution absorbance at ith wavelength.

### 2.5. Chlorophyll Fluorescence

The Fv/Fm of PSII (Photosystem II) was measured with the HandyPEA chlorophyll fluorescence system (Hansatech Instruments Ltd., Pentney, UK). Common duckweed leaves were placed in a leaf clip and stored in the dark for 30 min to quench chlorophyll fluorescence. After dark adaptation, chlorophyll was excited at light intensity of 2500 (µmol × m^−2^ × s^−1^), and minimum chlorophyll fluorescence (Fo), maximum chlorophyll fluorescence (Fm) and variable chlorophyll fluorescence (Fv = Fm − Fo) were determined. The Fv/Fm of PSII was determined based on the chlorophyll fluorescence kinetics of common duckweed.

### 2.6. Quinolone Biosorption

#### 2.6.1. Spectrophotometric Measurements

The extinction (*E*) of adsorbate solutions was measured with the Hitachi UV/VIS U-1800 spectrophotometer (Hitachi, Tokyo, Japan). The absorption spectra of QNs were set at 191–800 mm to determine the maximum absorption (*λ_max_*) of the analyzed solutions with a concentration of 1.6 mg × L^−1^ each. The *λ_max_* values for quantitative measurements were determined at: 210 and 333 nm for NAL, 274 nm for PEF, 290 nm for LVF and 287 nm for MOXI.

#### 2.6.2. Biosorption Measurements

The calibration curves of absorption vs. concentration *E* = *f(c)* were plotted for each adsorbate solution with the use of the previously determined *λ_max_* values. The Bs process was analyzed by placing 100 duckweed plants in 50 cm^3^ of adsorbate solution and stirring flask contents manually only once. The extinction of the analyzed solutions was determined before placing the plants in the solution (*E*_0_) and after 168 h (*E_7_*). After the measurement, 2 mL of the adsorbate solution was returned to the flask. The difference (C_0_ − C_7_) was the adsorption of a given adsorbate on the analyzed adsorbent.

Biosorption was calculated with the following formula:Bs = (C_0_ − C_7_)/m × V/1000(11)
where:

Bs—biosorption (mg × g^−1^);

C_0_—initial concentration of adsorbate (mg × L^−1^);

C_7_—concentration of adsorbate after 168 h (mg × L^−1^);

V—solution volume (mL);

m—adsorbent mass (mg).

### 2.7. Statistical Analysis

The experiment was conducted in six replicates. The results were expressed as means ± standard deviation (SD). Data were processed statistically by two-way analysis of variance—ANOVA (F test) (Table 2). The experimental factors were the type and concentration of the applied antibiotic. Significant differences were determined by Tukey’s test at *p* < 0.01. The results of the experiment (Ir, Iy, I_DW_, FM and DM of 100 duckweed plants, Chl *a* and Chl *b*, TCC, PQ, Fv/Fm and the Bs) of common duckweed were processed in the STATISTICA 13.3 statistical package (TIBCO Software Inc., Palo Alto, Santa Clara, CA, USA, 2018). The number and total area of fronds were used as the endpoints for assessing phytotoxicity [40,43,44]. Effective concentrations (EC_x_) were analyzed with a selected regression model to calculate the concentrations at 20% and 50% response levels.

## 3. Results

### 3.1. Effect of QNs on the Ir and Iy of Common Duckweed

The impact of increasing NAL, PEF, LVF and MOXI concentrations (0; 0.4; 0.8; 1.3; 3.2; 6.4; 12.8 mg × L^−1^) in the growth medium on the Ir, Iy, I_DW_, FM and DM, Chl *a*, Chl *b,* TCC, Fv/Fm and the PQ of common duckweed tissues was evaluated.

In common duckweed plants exposed to QNs, Ir and Iy were evaluated based on the number of fronds and frond area. The analyzed concentrations of QNs induced significant changes in the Ir and Iy of common duckweed (Table 2). After 7 days of exposure to the lowest concentration (0.4 mg × L^−1^) of NAL, PEF, LVF and MOXI, the Ir of duckweed plants was determined at 0.45, −18.81, 5.96 and 21.06%, respectively. The highest concentration of the tested drugs (12.8 mg × L^−1^) significantly increased Ir values by 58.47 and 81.57% in plants exposed to PEF and LVF, and insignificantly by 62.54 and 59.41% for NAL and MOXI (Figure 1A).

The yield of common duckweed plants decreased proportionally with a rise in QN concentrations in the medium, excluding 0.4 and 0.8 mg × L^−1^ of PEF. At the lowest of concentration of QNs (0.4 mg × L^−1^), the most toxic was MOXI (Iy = 27.81%), whereas at the highest concentration (12.8 mg × L^−1^), it was LVF (Iy = 81.78%) (Figure 1B).

The predicted toxic units were calculated based on QN content and EC values. The 7-day chronic toxicity test revealed that NAL, PEF, LVF and MOXI were toxic for common duckweed and increased the values of Ir and Iy when applied at concentrations higher than EC_20_ = 0.48 mg × L^−1^, EC_20_ = 1.45 mg × L^−1^, EC_20_ = 0.62 mg × L^−1^ and EC_20_ = 0.29 mg × L^−1^, respectively. NAL, PEF, LVF and MOXI applied at concentrations of 3.27 mg × L^−1^, 5.99 mg × L^−1^, 2.67 mg × L^−1^ and 3.98 mg × L^−1^ (EC_50_), respectively, increased the Ir and Iy of common duckweed by 50% (Table 3).

Exposure to QNs had an effect on the I_DW_ of common duckweed. The parameters that induced significant differences in the I_DW_ values of duckweed were the type and the concentration of the tested drugs (not for interaction of antibiotic’s type and concentration) (Table 2). Nalidixic acid was most phytotoxic for the DW of duckweed plants. The I_DW_ insignificantly increased to 13.89% already under exposure to the lowest concentration (0.4 mg × L^−1^) of NAL, whereas in plants exposed to PEF, the same (the highest for PEF) effect was observed only at a concentration of 3.2 mg × L^−1^. In plants treated with the highest concentration (12.8 mg × L^−1^) of NAL, I_DW_ values significantly reached 24.07%. Other drugs were less phytotoxic. Plants exposed to PEF, LVF and MOXI were characterized by insignificant I_DW_ of 8.97, 5.09 and 3.92%, respectively (Figure 1C). Nalidixic acid was the only drug that increased the I_DW_ of duckweed plants by 20% already at EC_20_ = 0.94 mg × L^−1^ (Table 3).

### 3.2. Fresh Mass and Dry Matter Content

Fresh mass (100 duckweed plants) was analyzed after 7 days of exposure to QNs. The type and concentrations of antibiotic significantly influenced the FM of duckweed (Table 2). The FM of control plants was 320.50 mg. Fresh mass insignificantly decreased already in response to the lowest antibiotic concentration (0.4 mg × L^−1^), excluding plants that were grown in the presence of LVF, where FM was somewhat higher at 324.33 mg.

Fresh mass continued to decrease higher concentrations of all QNs, significantly at NAL and MOXI, and it was determined at 241.67 and 272 mg under exposure to the highest (12.8 mg × L^−1^) concentration, respectively (Figure 2A). Nalidixic acid was the most toxic drug that decreased FM by 20% (EC_20_) when applied at a concentration of 1.09 mg × L^−1^, which was not observed under exposure to the remaining antibiotics (Table 3).

The DM of control plants was 5.15%. The type and concentrations of antibiotic significantly influenced the DM of duckweed (Table 2). Increasing concentrations of NAL, PEF, LVF promoted tissue dehydration and led to a non-significant but steady increase in DM values (Table 2). The increase in plant DM was observed under exposure to MOXI, which was the only QN that significantly increased DM up to 6.18% at a concentration of 0.85 mg × L^−1^, which corresponds to EC_20_ (Table 3). The average DM of plants treated with NAL, PEF and LVF reached 5.52% (Figure 2B).

### 3.3. Chlorophylls, Carotenoids, Phaeophytinization and Fluorescence

The leaves of control plants (grown in a medium without antibiotics) contained 1.26 mg × g^−1^ Chl *a* and 0.49 mg × g^−1^ Chl *b*, and their TCC was determined at 0.44 mg × g^−1^ of FM (Figure 3A–C). The content of all photosynthetic pigments was significantly modified by the type and concentration of tested drugs (Table 2). The content of Chl *a*, Chl *b* and TCC significantly decreased by 18% on average already in response to the lowest concentration of LVF (0.4 mg × L^−1^). The lowest concentrations of NAL, PEF and MOXI were somewhat less toxic for photosynthetic pigments. When the tested drugs were applied at the highest concentration (12.8 mg × L^−1^), the greatest significant reduction in the content of Chl *a*, Chl *b* and TCC (38%, 39% and 31%, respectively) was observed under the influence of MOXI. Nalidixic acid was least toxic by 14% on average for Chl *a*, Chl *b* and TCC and the highest concentration of NAL (12.8 mg × L^−1^) (Figure 3A–C). Chlorophyll *a* levels in plant tissues were reduced by 20% (EC_20_) in the presence of 0.63, 0.55, 0.61 and 0.58 mg × L^−1^ of NAL, PEF, LVF and MOXI, respectively, in the growth medium, as well as for the content of Chl *b* at 0.56, 0.50, 0.49, 0.57 mg × L^−1^, respectively. Moxifloxacin was the only drug that reduced TCC by 20% (EC_20_) at a concentration of 0.90 mg × L^−1^ (Table 3). None of the tested QNs influenced PQ values. PQ reached 1.23 on average in all samples (Table 2, Figure 3D).

The Fv/Fm ratio was affected by both the type and the concentration of the applied antibiotic (Table 2). Duckweed plants exposed to PEF were characterized by the lowest Fv/Fm ratio relative to control. The average value of Fv/Fm was 0.73. Moxifloxacin and NAL were less invasive (Fv/Fm of 0.76 on average). Among plants exposed to drugs, the Fv/Fm ratio was highest (on average 0.78) in plants grown in the presence of LVF. At the highest concentration of NAL, PEF, LVF and MOXI (12.8 mg × L^−1^), the Fv/Fm ratio was 6%, 10%, 4.5% and 7% insignificantly lower, respectively, than in the control (Figure 4).

### 3.4. Quinolone Biosorption

The uptake of NAL, MOXI, LVF by common duckweed during the 7-day chronic toxicity test was directly proportional to drug concentrations in the growth medium. Nalidixic acid was the most readily absorbed compound. Its content in common duckweed tissues was 0.72 mg × g^−1^ DM (statistically significant) at the lowest concentration of this drug (0.4 mg × L^−1^), and it reached 5.19 mg × g^−1^ DM at its highest concentration (12.8 mg × L^−1^). The biosorption of MOXI was somewhat lower, and it was determined at 0.16 (statistically insignificant) and 4.55 mg × g^−1^ DM (statistically significant) in plants exposed to 0.4 and 12.8 mg × L^−1^ of MOXI, respectively. Levofloxacin was not absorbed when supplied at the lowest concentration (0.4 mg × L^−1^).

The uptake of this drug began when LVF concentration was doubled (0.8 mg × L^−1^), and its uptake significantly increased in concentrations of LVF ≥ 3.2 mg × L^−1^. The LVF content of duckweed plants peaked at 1.31 mg × g^−1^ DM under exposure to 6.4 mg × L^−1^ LVF. In contrast, rising concentrations of PEF almost did not affect its biosorption, which was determined at 0.67 mg × g^−1^ DM on average in all samples (Figure 5).

## 4. Discussion

Plant growth and the development of morphological traits in plants are inhibited under exposure to pharmaceuticals that are present in soil and water [37,45,46,47,48]. The present study analyzed the effects of four QNS: NAL, PEF, LVF and MOXI, on duckweed yield, chlorophyll a (Chl *a*) and chlorophyll b (Chl *b*) content, total carotenoid content (TCC), phaeophytinization quotient (PQ), maximum quantum efficiency (Fv/Fm) and quinolone biosorption (Bs) by duckweed plants. The lowest concentration of PEF (0.4 mg × L^−1^) stimulated (by 17% on average) the Ir and Iy of common duckweed plants, whereas NAL, LVF and MOXI applied at this concentration were phytotoxic. Moxifloxacin was the strongest inhibitor of plant growth and yield, and it increased Ir and Iy values by 24% (Figure 1A,B). Low concentrations of pharmaceuticals in soil have a positive effect on plants and can even stimulate plant growth (hormesis occurs at low drug concentrations) [49]. However, the influence of pharmaceuticals on plants is determined not only by their concentration, the host species or exposure time, but also by the form of the drug. In the work of [48], a 1.25 mM dose of soluble ciprofloxacin was lethal for duckweed (Ir and Iy decreased by 100%), whereas insoluble ciprofloxacin did not induce such toxic effects even when applied at the highest dose (40 mM). In the current study, none of the tested QNs at any of the examined concentrations was lethal for common duckweed plants. However, when applied at the highest concentration (12.8 mg × L^−1^), LVF significantly increased Ir and Iy values by 82% on average, and NAL, PEF and MOXI by 62% on average (Figure 1A,B). Based on growth indicators alone, NAL is somewhat less toxic than LVF. However, QNs are not the only pharmaceuticals that can strongly inhibit the growth of common duckweed. The Ir and Iy of common duckweed were inhibited by 90% (EC_90_) by 40 µM of tetracycline, which is equivalent to 18 mg × L^−1^ of tetracycline [45]. Similarly to nearly all tested QN concentrations, other chemical compounds can also exert toxic effects on common duckweed. Sodium chloride and glyphosate herbicide inhibit plant growth, decrease FM and increase DM [50,51], and polyethylene microbeads affect root growth and reduce the viability of root cells [52]. A glyphosate concentration of 40 μM not only dehydrated common duckweed tissues but also induced nearly complete chlorosis and necrosis of fronds [51]. QNs also led to the loss of assimilation pigments. In the present study, the highest concentration (12.8 mg × L^−1^) of MOXI, a synthetic antibiotic, induced the greatest significant decrease (36% on average) in the content of assimilation pigments (Figure 3A–C). Meanwhile, in a study of thale cress (*Arabidopsis thaliana*), which is widely used as a model organism in biological research, the application of tetracycline, a natural antibiotic, did not induce a chloroplast-specific stress response because the expression of chloroplast-specific chaperones (PsaB and PsbA) was unaltered. Tetracycline is phytotoxic, as it induces the expression of mitochondrial stress genes, impairs mitochondrial translation and inhibits the function of mitochondria, which generate most of the cell’s supply of adenosine triphosphate (ATP) [53]. In this study, the tested concentrations of QNs had almost no effect on PQ values (Figure 3D), but they (excluding LVF ≤ 6.4 mg × L^−1^) induced changes in chlorophyll fluorescence. Exposure to PEF led to the greatest decrease in the Fv/Fm ratio (7.5% on average) relative to control. Moxifloxacin and NAL were somewhat less toxic, and they decreased the Fv/Fm ratio by 6.5% on average. Common duckweed plants responded completely differently to LVF. The average Fv/Fm ratio of plants grown in the presence of LVF was nearly identical to that noted in control plants (0.78) (Figure 4). Chlorophyll fluorescence is affected not only by pharmaceuticals. In duckweed plants exposed to glyphosate concentrations of 20 and 40 μM, the photochemical activity of photosystem II (PSII) (expressed as Fv/Fm) was inhibited in 62% and 95%, respectively [51]. Xenobiotics decrease the Fv/Fm ratio not only in lower vascular plants. The Fv/Fm values of Scots pine (*Pinus sylvestris* L.) needles and European beech (*Fagus sylvatica* L.) leaves decreased in soil contaminated with diesel oil. Scots pines adapt (grow more rapidly and produce higher biomass) to long-term soil contamination with diesel [54]. In the current study, duckweed plants also adapted to low concentrations of PEF (0.4–0.8 mg × L^−1^). However, their growth was inhibited by 20% (EC_20_) in response to a PEF concentration of 1.45 mg × L^−1^ (Figure 1A,B; Table 3). The lowest observed effect concentration (LOEC) is the concentration at which a chemical compound reduces the measured response by more than 20% [55]. The LOEC values of NAL, LVF and MOXI, which inhibited the growth of duckweed plants and decreased their yield (mean Ir and Iy), were determined at >0.48, >0.62 and >0.29 mg × L^−1^, respectively (Table 3). According to [56], tetracycline and chlortetracycline inhibited the growth of lettuce (*Lactuca sativa* L.) to a different degree. Chlortetracycline exerted a bi-directional effect on plant growth, where low concentrations of the drug promoted growth, whereas high concentrations inhibited growth and photosynthesis. While the common duckweed accumulates tetracycline, it accumulates biogenic amines such as: tyramine, putrescine, cadaverine, spermidine and spermine. The amines spermine and tyramine were most sensitive when the plants were treated with this drug [45]. Common duckweed plants also react with various changes as a result of exposure to substances other than drugs. These plants exposed to glyphosate also accumulate biogenic amines, but mainly due to an increase in the activity of tyrosine decarboxylase and ornithine decarboxylase: enzymes of the biogenic amine biosynthetic pathway [57]. In addition, changes in the activity of antioxidant enzymes are characteristic when these plants have disturbed homeostasis. They respond to glyphosate by increasing peroxidase activity [57], and exposure to silver nitrate corresponds to an increase in catalase activity and content of malondialdehyde, and these answers and morphological parameters may be different for *Lemna minor* and *Lemna minuta* [58].

The biosorption of the tested drugs from the growth medium compromised the morphological and physiological parameters of common duckweed. Nalidixic acid was most readily absorbed, and its uptake significantly reached 0.72 mg × g^−1^ DM at the lowest concentration (0.4 mg × L^−1^) and 5.19 mg × g^−1^ DM at the highest concentration (12.8 mg × L^−1^) (Figure 5). Nalidixic acid was the most toxic drug that decreased I_DW_ and FM by 20% (EC_20_) at concentration of 1.02 mg × L^−1^ on average, which was not observed under exposure to the remaining antibiotics (Table 3). However, an analysis of plant physiological parameters, including the content of assimilation pigments, revealed that PEF was more toxic for the studied plants (Figure 2B and Figure 3A,C). According to Gadd [59], biosorption occurs independently of physicochemical metabolism during which substances are removed from a solution by biological materials. Biosorption consists of a solid phase (biosorbent) and a liquid phase (solvent, usually water) containing dissolved or suspended substances that are sorbed (sorbate) [38]. Nalidixic acid has the lowest molar mass [60] in the group of the tested drugs, and it was most effectively removed from the solution by duckweed plants. The molar mass of MOXI is nearly twice higher in comparison with NAL [61], but this drug was sorbed 22% less effectively than NAL on average. The uptake of all tested FQNs (MOXI, LVF and PEF) was generally lower than NAL uptake (Figure 5). However, biological materials show affinity for inorganic and organic contaminants, which indicates that these matrices have a high potential for absorbing various types of substances (Gadd, 2009). The uptake of drugs inhibits the development of morphological parameters in plants [48]. The presence of MOXI in a solution retarded the growth of *Pseudokirchneriella subcapitata* algae [62], whereas *Chlorella vulgaris* algae in a highly saline environment biodegraded LVF, minimized the drug’s ecotoxicity and decreased the resistance of bacteria targeted by this antibiotic [63]. Therefore, the identification of the most effective, available and inexpensive biomaterials poses the greatest challenge in biosorption research. Living organisms with biosorption capacity should be widely available, abundant in nature and easy to cultivate [64]. Common duckweed is an indicator plant that meets the above requirements [40,43].

## 5. Conclusions

The results of this study confirm that the Lemna test is a useful analytical tool for evaluating the toxicity of QNs and for predicting the consequences of the chemical contamination of freshwater bodies based on the calculated values of toxicity indicators. The presence of the tested QNs in water triggered negative morphological and physiological responses in duckweed, which suggests that these compounds not only decrease its populations but also impact other trophic levels in the aquatic food webs.

In the current study, none of the tested QNs at any of the examined concentrations were lethal for common duckweed plants. However, at the highest concentration (12.8 mg × L^−1^), LVF increased Ir and Iy values by 82% on average, and NAL, PEF and MOXI by 62% on average. The uptake of NAL, MOXI, LVF by common duckweed during the 7-day chronic toxicity test was directly proportional to drug concentrations in the growth medium. Nalidixic acid was absorbed in the largest quantities, whereas in the group of FQNs, MOXI, LVF and PEF were less effectively absorbed by common duckweed. Common duckweed removes QNs from the solution by active rather than by passive biosorption (dead biomass), and it can transform the accumulated pollutants by internal processes, thus reducing their concentrations in the environment.

The research demonstrated that biosorption by common duckweed occurs regardless of the plants’ condition. In addition, plants that absorb contaminating substances can be easily removed from the solution and neutralized. Even compromised plants removed QNs from the growth medium to a varied extent. These findings indicate that common duckweed can be used as an effective biological method to remove QNs from wastewater and water, and that biosorption should be a mandatory process in conventional water and wastewater treatment.

## Figures and Tables

**Figure 1 ijerph-20-05089-f001:**
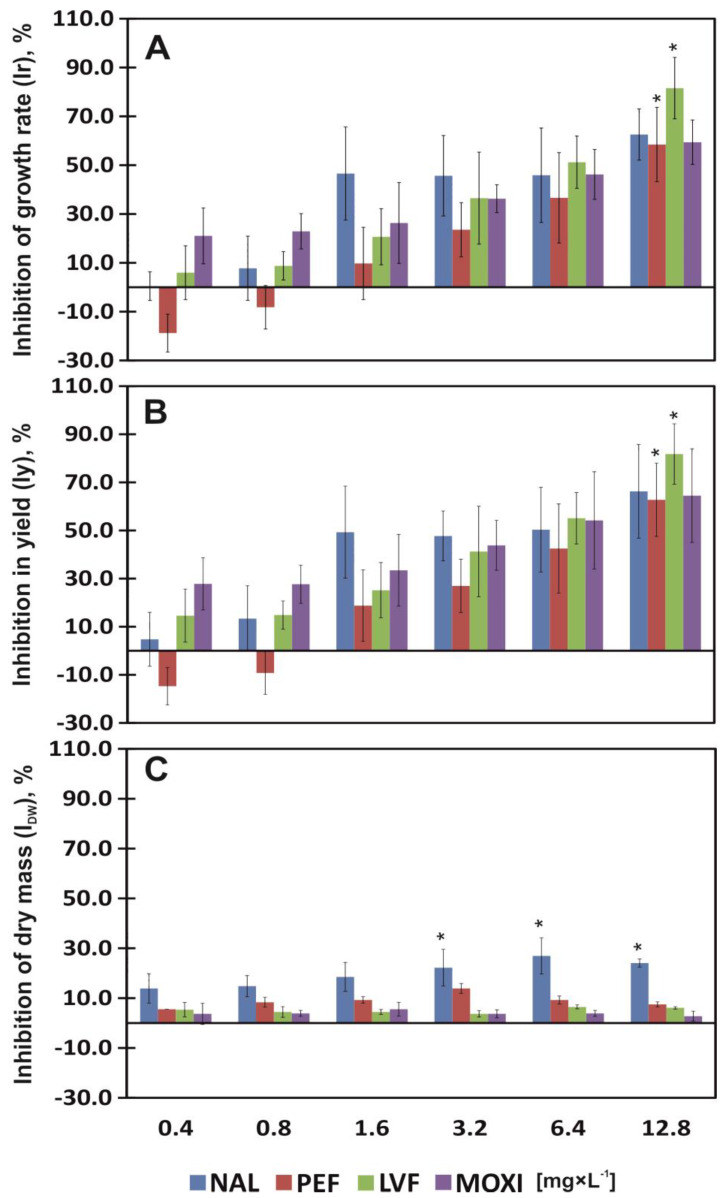
Percent inhibition of growth rate (Ir) (**A**), percent reduction in yield (Iy) (**B**) and percent reduction in the dry weight (I_DW_) (**C**) of common duckweed (*L. minor*) exposed to different concentrations (0–12.8 mg × L^−1^) of nalidixic acid (NAL), pefloxacin mesylate dihydrate (PEF), levofloxacin (LVF) or moxifloxacin hydrochloride (MOXI). Data points represent the mean ± SD, n = 6. * Values differ significantly from the control at *p* < 0.01.

**Figure 2 ijerph-20-05089-f002:**
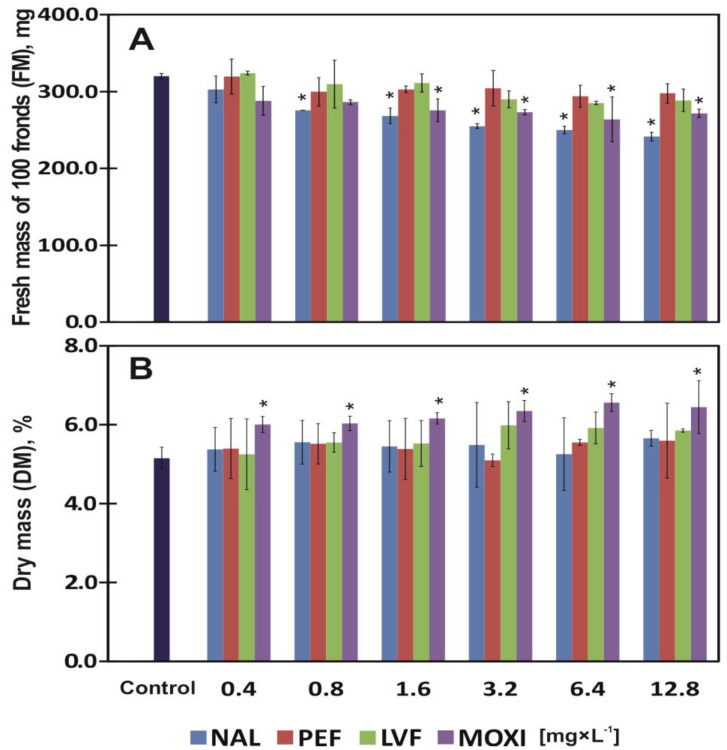
Fresh mass (FM) (**A**) of 100 fronds and dry matter content (DM) (**B**) of common duckweed (*L. minor*) exposed to different concentrations of (0–12.8 mg × L^−1^) nalidixic acid (NAL), pefloxacin mesylate dihydrate (PEF), levofloxacin (LVF) or moxifloxacin hydrochloride (MOXI). Data points represent the mean ± SD, n = 6. * Values differ significantly from the control at *p* < 0.01.

**Figure 3 ijerph-20-05089-f003:**
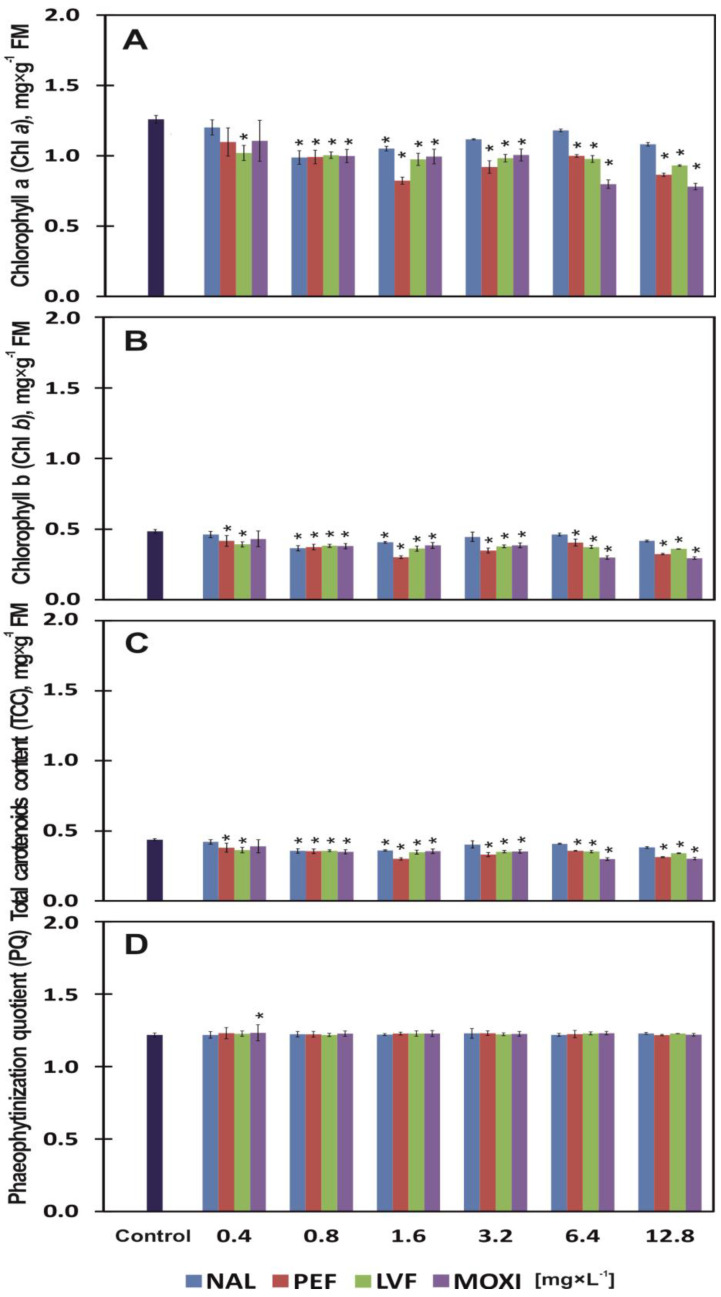
Content of chlorophyll *a* (Chl *a*) (**A**), chlorophyll *b* (Chl *b*) (**B**), total carotenoid content (TCC) (**C**), and the phaeophytinization quotient (PQ) (**D**) of common duckweed (*L. minor*) exposed to different concentrations (0–12.8 mg × L ^−1^) of nalidixic acid (NAL), pefloxacin mesylate dihydrate (PEF), levofloxacin (LVF) or moxifloxacin hydrochloride (MOXI). Data points represent the mean ± SD, n = 6. * Values differ significantly from the control at *p* < 0.01.

**Figure 4 ijerph-20-05089-f004:**
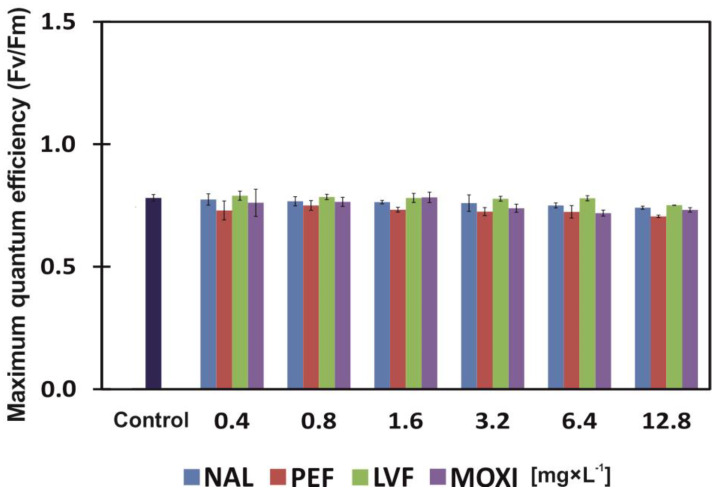
Maximum quantum efficiency (Fv/Fm) of common duckweed (*L. minor*) exposed to different concentrations (0–12.8 mg × L^−1^) of nalidixic acid (NAL), pefloxacin mesylate dihydrate (PEF), levofloxacin (LVF) or moxifloxacin hydrochloride (MOXI). Data points represent the mean ± SD, n = 6.

**Figure 5 ijerph-20-05089-f005:**
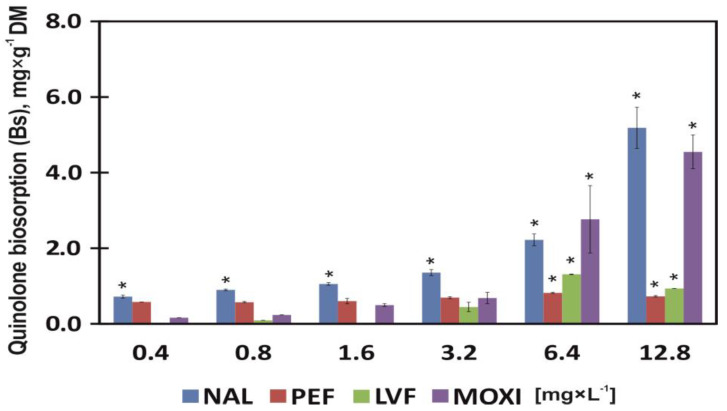
Quinolone biosorption by common duckweed (*L. minor*) exposed to different concentrations (0–12.8 mg × L^−1^) of nalidixic acid (NAL), pefloxacin mesylate dihydrate (PEF), levofloxacin (LVF) or moxifloxacin hydrochloride (MOXI). Data points represent the mean ± SD, n = 6. * Values differ significantly from the control at *p* < 0.01.

**Table 1 ijerph-20-05089-t001:** Characteristics of the QNs were used to prepare the test concentrations in the experiment.

Chemical Compound	Structural Formula	Empirical Formula	CAS Number	Molecular Weight (g × moL^−1^)	Form/Color
**Nalidixic acid** **(NAL)**	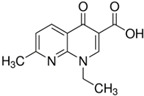	C_12_H_12_N_2_O_3_	389-08-2	232.24	powder/ beige
**Pefloxacin ** **mesylate ** **dihydrate** **(PEF)**	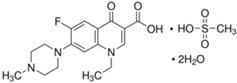	C_18_H_28_FN_3_O_8_S	149676-40-4	465.49	powder/ white or almost white
**Levofloxacin** **(LVF)**	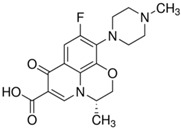	C_18_H_20_FN_3_O_4_	100986-85-4	361.37	powder/ light yellow
**Moxifloxacin ** **hydrochloride** **(MOXI)**	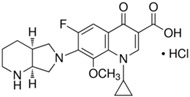	C_21_H_24_FN_3_O_4_ · HCl	186826-86-8	437.89	crystalline/ colorless

**Table 2 ijerph-20-05089-t002:** Analysis of variance (ANOVA) of the morphological and biochemical parameters (and indices) of common duckweed (*L. minor* L.).

SoV	Ir	Iy	IDW	FM	DM	Chl a	Chl b	TCC	PQ	Fv/Fm	Bs
	F-value										
Intercept	87.48 *	121.81 *	137.12 *	52,356.81 *	10,574.14 *	29,592.25 *	27,794.50 *	37,283.73 *	12,050,801 *	103,107.30 *	1609.11 *
Antibiotic (A)	1.76	2.22	20.54 *	39.09 *	9.19 *	29.49 *	29.52 *	25.56 *	4.00 *	16.70 *	144.85 *
Concentration (C)	11.68 *	12.41 *	4.55 *	27.09 *	2.69 *	52.11 *	53.23 *	49.61 *	7.00 *	7.50 *	243.71 *
AxC	0.50	0.38	1.08	2.24 *	0.61	5.83 *	7.45 *	4.66 *	6.00 *	0.90	45.70 *

SoV—source of variation, I—intercept A—type of antibiotic, C—concentration, A × C—antibiotic × concentration interactions, * significant at *p* < 0.01.

**Table 3 ijerph-20-05089-t003:** The effect of nalidixic acid (NAL), pefloxacin mesylate dihydrate (PEF), levofloxacin (LVF) and/or moxifloxacin hydrochloride (MOXI) on plant parameters: percent inhibition of growth rate (Ir), percent reduction in yield (Iy), percent reduction in dry weight (I_DW_), chlorophyll *a* (Chl *a*) and *b* (Chl *b*), total carotenoid content (TCC), phaeophytinization quotient (PQ) of tissues, maximum quantum efficiency (Fv/Fm), fresh mass (FM) and dry matter content (DM) of 100 duckweed plants.

Antibiotic	Parameter	Effective Concentration, mg × L^−1^	
		EC_20_	EC_50_
NAL	Ir	0.53	3.69
	Iy	0.42	2.85
	Mean Ir and Iy	0.48	3.27
	I_DW_	0.94	-
	FM	1.09	-
	DM	-	-
	Chl *a*	0.63	-
	Chl *b*	0.56	-
	TCC	-	-
	PQ	-	-
	Fv/Fm	-	-
PEF	Ir	1.59	6.67
	Iy	1.31	5.30
	Mean Ir and Iy	1.45	5.99
	I_DW_	-	-
	FM	-	-
	DM	-	-
	Chl *a*	0.55	-
	Chl *b*	0.50	-
	TCC	0.55	-
	PQ	-	-
	Fv/Fm	-	-
LVF	Ir	0.71	2.89
	Iy	0.53	2.45
	Mean Ir and Iy	0.62	2.67
	I_DW_	-	-
	FM	-	-
	DM	-	-
	Chl *a*	0.61	-
	Chl *b*	0.49	-
	TCC	0.70	-
	PQ	-	-
	Fv/Fm	-	-
MOXI	Ir	0.36	5.10
	Iy	0.21	2.86
	Mean Ir and Iy	0.29	3.98
	I_DW_	-	-
	FM	-	-
	DM	0.85	-
	Chl *a*	0.58	-
	Chl *b*	0.57	-
	TCC	0.93	-
	PQ	-	-
	Fv/Fm	-	-

## Data Availability

Not applicable.

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
