# Peer review of "The Effect of Quinolones on Common Duckweed Lemna minor L., a Hydrophyte Bioindicator of Environmental Pollution"

_ijerph, 2023, doi:10.3390/ijerph20065089_

Round 1

Reviewer 1 Report

1. The introduction should include the functional characteristics of aquatic plants

2. line 102 Basic information of species should be provided;

3. line 114 How is the concentration setting referenced?

4. line 387 More cases should be added about the response characteristics of duckweed to pollutants (10.1016/j.envpol.2021.116900; 10.1002/jat.2997)

5. line 424 The response process of aquatic plants is very important, mainly reflected in morphological and physiological characteristics, so it should be discussed in depth (10.1016/j.aquabot.2011.07.001;

10.1016/j.envint.2021.106708;

10.1016/j.aquatox.2022.106260)

Author Response

Dear Reviewer,

we appreciate your interest that You have taken in our manuscript and the constructive criticism You have given. We have addressed the major concerns contained in your Review. In attached file find  a description/discussion of how we have addressed your comments.

Best regards

Łukasz Sikorski

Agnieszka Bęś

Kazimierz Warmiński

Reviewer 2 Report

The aim of this study was to determine the effects of exposure to four antibiotics. The phytotoxicity of the antimicrobials was determined by evaluating the morphological and physiological parameters of common duckweed with the aim of establishing common duckweed’ potential as a tool for removing these pharmaceuticals from wastewater and water.

In all the paper is well crafted and needs slight revision.

 The was no section on quality control: how reliable is the data obtained

Author Response

(The authors gave the same response as above.)

Reviewer 3 Report

The manuscript deals with the morphological and physiological characteristics of duckweed in response to antibiotics exposure. Overall, it is an interesting study. However, there are certain things that need to be addressed.

-What is the novelty of the present work? It should be mentioned in the last paragraph of introduction.

-Last paragraph of introduction is more methodology and results oriented, these parts must be moved to the concerned sections.

-Is it possible to quantify the antibiotics by any other technique instead of spectrophotometery? or have you compared with any other advance technique? 

-Figure 1, the maximum value on y-axis should be 100% that will improve the comparison and visibility. 

-Figure 3. y-axis should be same for all the parts, e.g., it is upto 3 in fig 3A and 2 in others. 

-All the figures, decimals must be upto the same number, i.e. 1 or 2. These should be like 1.0, 1.5, 2.0, .... instead of 1, 1.5, 2, ... on the y-axis. 

-Botanical/Technical names should be italicised.

-Conclusion is very long. It should be focused and to the point.

-Based upon the outcomes of the present study, what are possible future perspectives? Please mention in conclusion part. 

 -There are many formatting mistakes that must be corrected. 

Author Response

(The authors gave the same response as above.)

Reviewer 4 Report

The reviewed article provides insight into the toxicity of quinolone antibiotics (QNs) to the common duck Lemna minor L., as well as evaluates the potential of L. minor to remove QNs from the aquatic environment. By Scopus search with phrases lemna  AND minor  AND quinolones only three outputs have been found, therefore the article reflects some novelty. I have found that the following paper reflects similar concepts https://doi.org/10.1016/j.etap.2019.103242, therefore I recommend applying it for introduction or discussion section to emphasize the novelty of the current study. The reviewed article meets the scope of the IJERPH journal, as well represent overall good quality. However, I have identified some several issues that should be solved before the final acceptance.

Major comments

1. The title should be modified. The current version emphasises the removal of QNs, while the abstract, as well as much of the manuscript, focuses on the biological response to QNs treated as an intoxicant. This is especially true in the context of the stated aim of the study. There is no mention of removal. The ability to remove is not given enough prominence in the main text.

2. The introduction needs a short revision and better organization of the material. It starts off on the right foot i.e. pointing out the issues related to the release of xenobiotics into the environment and then the defined role of antibiotics is presented. Up to this point it is good.

2a) Lines 45-37 state that antibiotics cause significant risks to human health, please provide specific examples and relate these to specific doses. Similarly in lines 48-50, what biological effect can antibiotics concentrations in ng/L have?

2b) What is the point to mention the list of the analysed antibiotics in the middle of the section. First introduce to QNs and present functional groups. Then specify analysed cases along with the aim of the study.

2c) In my opinion, especially in relation to the current manuscript title, the technologies and processes responsible for antibiotics removal from the wastewater are not sufficiently exposed.  Please add.

2d) finally decide what is the actual scientific goal and relate this to the title: removal of antibiotics from the aquatic environment or their toxic effect of on L. minor. 

3. Materials and methods

3a) Provide more details about applied plants, any code for the culture? What are the conditions for cultivation at the standard conditions? Do Lemna is prepared somehow for the toxicity tests?

3b) Lines 108-109 The reagents were supplied in the form of the standardized solutions? Please specify, what were their concentrations or how the solutions for the experiment were designed. 

3c) I guess that it is better to provide explanations to abbrev. for TCC, PQ, Fv/Fm in the M&M section rather than in introduction. 

3d) If the removal of QNs was previously considered as a main topic, the ratio of the concentration of antibiotics removed by biosorption to the concentration remaining in solution should also be assessed.

4. Results and discussion.

4a) The results should be described in more synthetic way, in particular section 3.3 needs some shortening and to emphasize the key trends, instead of very detailed description of each output. 

4b) I suggest enlarging or minimize scales in Figure 3 to better emphasize the differences for the results between control and experimental groups. 

In the discussion section, the benefits of L. minor application as a maker organism or inducer of antibiotics/xenobiotics removal over currently applied plants should be expanded and specified. 

5. Conclusion section in great parts repeat information from the abstract and even got the same sentences. More efforts have to be done to shorten and expose key findings obtained during the studies. To be honest, sentences/ideas from lines 451-457 fits greatly for conclusions part.

6. Careful English proofed, has to be performed before the final publication. 

Minor comments:

I. This sentence from introduction and conclusion is controversial: Plant growth and the development of morphological traits in plants are often inhibited under exposure to most pharmaceuticals that are present in soil and water. I guess not often but at specified concentrations level. Such, formulation is highly inaccurate, I suggest rephrasing. 

II. Fluoroquinolones (FQNs) abbreviation should be explained when first occur in the text (line 63 instead of 65).

Author Response

(The authors gave the same response as above.)

Round 2

Reviewer 4 Report

The Authors have responded satisfactorily to all my comments.  After provided modifications and final proofread, the manuscript is ready for publication.